# Recent Advances in Microfluidic Impedance Detection: Principle, Design and Applications

**DOI:** 10.3390/mi16060683

**Published:** 2025-06-05

**Authors:** Yigang Shen, Zhenxiao Wang, Tingyu Ren, Jianming Wen, Jianping Li, Tao Tang

**Affiliations:** 1The Institute of Precision Machinery and Smart Structure, College of Engineering, Zhejiang Normal University, Jinhua 321004, China; yigangshen@zjnu.edu.cn (Y.S.); 1029502@zjnu.edu.cn (Z.W.); rentingyu001205@163.com (T.R.); wjming@zjnu.cn (J.W.); 2College of Mathematical Medicine, Zhejiang Normal University, Jinhua 321004, China; 3Department of Neurosurgery, Chongqing General Hospital, Chongqing University, Chongqing 401147, China

**Keywords:** microfluidics, impedance, cell equivalent circuits, single cell

## Abstract

Under the dual drivers of precision medicine development and health monitoring demands, the development of real-time biosensing technologies has emerged as a key breakthrough in the field of life science analytics. Microfluidic impedance detection technology, achieved through the integration of microscale fluid manipulation and bioimpedance spectrum analysis, has enabled the real-time monitoring of biological samples ranging from single cells to organ-level systems, now standing at the forefront of biological real-time detection research. This review systematically summarizes the core principles of microfluidic impedance detection technology, modeling methods for cell equivalent circuits, system optimization strategies, and recent research advancements in biological detection applications. We first elucidate the fundamental principles of microfluidic impedance detection technologies, followed by a comprehensive analysis of cellular equivalent circuit model construction and microfluidic system design optimization strategies. Subsequently, we categorize applications based on biological sample types, elaborating on respective research progress and existing challenges. This review concludes with prospective insights into future developmental trajectories. We hope this work will provide novel research perspectives for advancing microfluidic impedance detection technology while stimulating interdisciplinary collaboration among researchers in biology, medicine, chemistry, and physics to propel technological innovation collectively.

## 1. Introduction

Amid frequent global health crises (e.g., COVID-19 pandemic, dengue fever outbreaks), surging demands for precision medicine, and escalating environmental and food safety challenges, the development of rapid, sensitive, and portable detection technologies has become a shared pursuit across scientific and industrial communities. Traditional detection methods (such as polymerase chain reaction (PCR) [1] and enzyme-linked immunosorbent assay (ELISA) [2]), while characterized by high specificity, are constrained by operational complexity, expensive equipment, and time-consuming procedures, making them inadequate for urgent on-site analysis requirements. As a groundbreaking solution, point-of-care testing (POCT) technology enables direct detection at sampling sites through portable devices and integrated reagents, eliminating the need for complex laboratory processing and facilitating the immediate acquisition of test results. This technology aims to enhance medical decision-making efficiency through device portability and streamlined workflows that rapidly deliver analytical outcomes.

Microfluidic technology is an advanced methodology that enables the precise manipulation of nano- to pico-liter scale fluids through engineered microscale channels and structures (typically tens to hundreds of micrometers). Its defining characteristic lies in integrating conventional laboratory functions into miniaturized chips, earning the designation “lab on a chip”. The concept of microfluidics was first proposed by Eringen et al. in 1964 [3]. In 1979, Terry et al. [4] developed a silicon-based gas chromatographic device that served as the prototype for modern microfluidic systems. Manz et al. introduced the conceptual framework for microfluidic design in 1990 [5]. A milestone advancement occurred in 1998 when the Whitesides group at Harvard University, through iterative material testing and based on microfabrication techniques, innovatively proposed using polydimethylsiloxane (PDMS) and simplified manufacturing methods to fabricate microfluidic chips [6], thereby establishing a comprehensive technical system. Through synergistic innovations in micro/nanofabrication processes (e.g., photolithography, soft lithography) and novel materials (e.g., PDMS, glass), microfluidic technology has evolved into an interdisciplinary field combining fluid dynamics, biomedicine, and sensing technologies. Its core advantages manifest in four dimensions: miniaturization (reducing sample/reagent consumption to 1/1000 of conventional methods), high-throughput capacity (enabling the parallel processing of thousands of reaction units), precise control (regulating particle motion via multi-physical fields including electric and pressure fields), and system integration (achieving unified sample pretreatment, reaction, and detection modules) [7,8,9,10].

Impedance detection technology is a universal analytical methodology based on measurements of impedance characteristics in circuits or materials under alternating current excitation. By analyzing the variations in impedance magnitude and phase with frequency, this technique reveals the electrical, physical, and chemical properties of analyzed subjects. In 1975, Kivelson et al. [11] advanced a theoretical model elucidating the relationship between polarization effects and frequency through investigations of dielectric relaxation behaviors in complex systems, providing fundamental theoretical support for the application of alternating current (AC) impedance spectroscopy in heterogeneous material analysis. A pivotal milestone emerged in 1999 when Ayliffe et al. [12] pioneered single-cell impedance measurements within microchannels, utilizing paired microelectrodes with a few-micrometer gap to achieve the impedance detection of human polymorphonuclear leukocytes and fish erythrocytes in 10 μm wide microchannels, thereby demonstrating the first evidence of single-cell electrical property discrimination capabilities. This technology exhibits three core advantages: non-invasiveness (sample preservation), multidimensional information (a comprehensive analysis of parameters including the impedance modulus, phase angle, and spectral curves), and high sensitivity (detectable to minute variations) [13,14].

The deep integration of microfluidics with impedance detection technologies has established revolutionary pathways for point-of-care testing and on-site analysis. This synergistic convergence not only enhances analytical dimensionality and precision but also drives a paradigm shift from laboratory-based to field-deployable platforms, achieving breakthroughs from macroscopic solutions to the microscale detection of single cells and bacteria. The defining features of microfluidic impedance detection technology encompass label-free operation (eliminating fluorescent/radioactive labeling), real-time dynamic monitoring (millisecond-level responsiveness), multiparameter resolution (membrane capacitance, cytoplasmic conductivity, and dielectric constant), and ultrahigh sensitivity (detection capability down to single microorganisms or microscale particles). Currently, three primary application matrices are established: fundamental research (single-cell heterogeneity analysis and organ-on-a-chip development), clinical diagnostics (rapid pathogen screening and bedside testing), and industrial applications (high-throughput drug screening and real-time environmental pollutant monitoring).

This review systematically examines the technical principles, system design methodologies, and recent advancements in microfluidic impedance sensing technology, with particular emphasis on its critical research value in biological detection, spanning from single-cell to organ-level analyses. A rigorous analytical framework is established to dissect the modeling approaches for cellular equivalent circuit models in microfluidic impedance systems, followed by comprehensive discussions on design principles categorized under static and dynamic operational paradigms. From an application perspective, the current implementation domains of microfluidic impedance analysis technology are classified according to biological sample types, delineated into four major categories, tumor detection, blood detection, organ on a chip, and microbial detection, as shown in Figure 1. Finally, this review culminates in a critical examination of prevailing technical bottlenecks (e.g., signal noise suppression, multimodal data fusion) and prospective breakthrough directions (e.g., AI-driven intelligent algorithms, flexible electronics integration), aiming to provide theoretical foundations and technical roadmaps for interdisciplinary innovation in precision medicine, green chemistry, and POCT fields.

## 2. Theory of Microfluidic Impedance Analysis

This chapter establishes a theoretical framework for microfluidic impedance analysis, spanning three domains: hydrodynamics, cellular equivalent circuit modeling, and system design. Hydrodynamic characteristics and microchannel flow behaviors are first elucidated. Equivalent circuit modeling methods for detecting cellular electro-mechanical properties via impedance measurements are subsequently introduced. Design principles for static and dynamic microfluidic impedance detection systems are comprehensively detailed, encompassing electrode configurations and accuracy enhancement strategies through electric field optimization, hydrodynamic focusing, and signal compensation. Progressing systematically from microscopic mechanisms to macroscopic implementations, this hierarchical framework provides the technical foundation for multidimensional single-cell characterization and precision sensing.

### 2.1. Microchannel Flow Characterization

The essence of microfluidics lies in constructing miniaturized fluidic networks, where microscale channels (specifically width, height, and depth dimensions within the micrometer range) are precisely fabricated via microfabrication techniques (e.g., soft lithography, laser ablation, or injection molding). Multi-physical field coupling effects (pressure, electric fields, and capillary forces) are harnessed to achieve controlled fluid delivery, mixing, separation, and reaction control. Microfluidic chips are typically fabricated using materials such as PDMS, glass, or thermoplastic polymers. Channel wettability is adjusted by surface chemical modifications (e.g., silanization or plasma treatment) to suit different application requirements. Since the dimensions of microfluidic channels are sub-micron to millimeter, the hydrodynamic behavior within these confined spaces is mainly governed by microscale effects, exhibiting four characteristic phenomena: under low Reynolds number conditions, laminar flow dominates, and interlayer material exchange is achieved only by molecular diffusion; due to the high surface area-to-volume ratio, the effects of surface tension, wettability, and wall adsorption are significantly enhanced; at the microscale, synergistic interactions between electric, thermal, and hydrodynamic fields are dramatically enhanced; and the shortened diffusion path and improved mass/heat transfer efficiency in microliter-scale volumes reduce reagent consumption to 1/100–1/1000 of conventional methods.

### 2.2. Cell Model

The impedance testing technique is a general analytical method that quantifies the internal electrical characteristics and structural features of a test object by measuring the impedance response under alternating electric field excitation in combination with spectral analysis and equivalent circuit modeling. The principle of operation involves applying a frequency-controlled AC excitation signal to the test object, collecting voltage and current waveform data synchronously, and calculating the amplitude ratio and phase difference to construct a frequency-dependent complex impedance spectrum. Subsequently, RLC equivalent circuit modeling or dielectric relaxation theory can establish quantitative correlations between characteristic frequency peaks, capacitive arc radii in the impedance spectrum, and physical/chemical properties (e.g., dielectric constant, conductivity). Impedance detection, characterized by non-invasiveness, high sensitivity, and cost-effectiveness, has become an important tool for linking microscopic biophysical and electrical properties to macroscopic applications in medicine, environmental monitoring, and food safety.

Impedance detection techniques are used in the electrical characterization of cells that can be represented as a uniformly conducting cytoplasm (cytoplasmic conductivity σ_i_) wrapped by a dielectric membrane (specific membrane capacitance C_sm_) according to the widely used spherical monoshell model [15]. When suspended cells are exposed to an AC electric field, the electrical properties of the cell–medium mixture can be represented by Maxwell’s mixture theory (MMT) and the corresponding equivalent circuit model (ECM), as shown in Figure 2a. In the simplest analytical model, the impedance consists of solution resistance (R_med_), solution capacitance (C_med_), cytoplasmic resistance (R_i_), membrane capacitance(C_mem_), double-layer capacitance (C_DL_), and stray capacitance (C_s_).(1)$$Z_{\mathrm{mix}\;}^{*}=\frac{R_{\mathrm{med}\;}\left(1+j\omega R_{i}C_{\mathrm{mem}\;}\right)}{j\omega R_{\mathrm{med}\;}C_{\mathrm{mem}\;}+\left(1+j\omega R_{i}C_{\mathrm{mem}\;}\right)\left(1+j\omega R_{\mathrm{med}\;}C_{\mathrm{med}\;}\right)}$$

Considering C_s_ and C_DL_, the total impedance is given by(2)$$Z_{\mathrm{total}\;}^{*}=\frac{2Z_{\mathrm{mix}\;}^{*}}{2j\omega C_{s}Z_{\mathrm{mix}\;}^{*}+j\omega C_{DL}Z_{\mathrm{mix}\;}^{*}+2}$$

Cellular impedance measurements leverage frequency-dependent characteristics: lower frequencies (<0.1 MHz) primarily analyze cell size, membrane properties, and concentration quantification, while higher frequencies (>0.1 MHz) probe intracellular features including organelle status, cytoplasmic composition, and pathophysiological conditions. At different frequencies, the main factors affecting impedance during cell impedance measurements vary, as shown in Figure 2c. At the first frequency dispersion (<0.1 MHz), the presence of the dielectric membrane causes the impedance to be primarily determined by the electric double-layer capacitance and cell size; at the second frequency dispersion (0.1–0.5 MHz), the electric field penetrates the membrane, and the impedance is predominantly determined by its dielectric constant; at the third dispersion (0.5–10 MHz), the dielectric membrane becomes permeable to the electric field, and the impedance is strongly influenced by intracellular conductivity (i.e., the ability of ions or electrons to migrate within the cytoplasm); and at the fourth dispersion (>10 MHz), the impedance is affected by stray capacitance [16]. Intrinsic cellular electrical properties are typically resolved by minimizing the squared discrepancy between measured impedance Z*_exp_ and model-estimated Z*_total_:(3)$$min\sum_{n}{\left[Z_{\exp\;}^{*}\left(\omega_{n}\right)-Z_{\mathrm{total}\;}^{*}\left(\omega_{n}\right)\right]}^{2},n=1,2,\ldots ,N$$

### 2.3. Cell Deformation Model

In the mechanical characterization of cells using impedance detection technology, resolving intrinsic mechanical parameters necessitates mapping the power-law rheological deformation process of single-cell viscoelasticity to corresponding impedance signals. Within the mechanical sensing zone, where single cells displace equivalent-volume media, an electrical equivalent model can be constructed based on sensing geometry, as shown in Figure 2b. Excitation frequencies are selected within the intermediate range (0.5–10 MHz) to eliminate C_s_ and C_DL_ influences. Consequently, the total resistance in the mechanical sensing zone is expressed as(4)$$R_{\mathrm{overall}\;}=R_{\mathrm{cell}\;}+R_{\mathrm{med}\;}=\frac{\rho_{\mathrm{med}\;}L_{\mathrm{channel}\;}}{S}+\frac{\left(\rho_{\mathrm{cell}\;}-\rho_{\mathrm{med}\;}\right)}{S}\cdot L_{p}$$
where ρ denotes resistivity (cell or medium), L represents protruding cell length or medium path in constriction channels (L_channel_ = L_p_ + L_med_), and S indicates the channel cross-sectional area. The measured resistance R_overall_ thus proportionally correlates with cellular protrusion length L_p_ under constant L_channel_, enabling the real-time tracking of dynamic deformation processes.

Cellular deformation derived from impedance measurements is subsequently utilized to resolve intrinsic mechanical parameters. According to power-law rheology [17], the cell undergoes three distinct phases during its passage through the constriction channel, as shown in Figure 2d. Phase I begins as the cell rapidly enters the constriction channel. Phase II involves the elastic deformation of the cell within the channel, exhibiting power-law deformation behavior consistent with the rheological model; this phase allows the determination of intrinsic mechanical parameters. Phase III occurs when the cell is fully deformed within the constriction channel and accelerates through it.

During Phase II, cellular viscoelastic deformation occurs within the microchannel, where the measured deformation length L_p_(t) adheres to a power-law rheological model. This dynamic behavior is governed by intrinsic cellular viscoelasticity, with core parameters including Young’s modulus and the power-law exponent. Subsequently, deformation data L_p_(t) from Phase II is utilized to generate theoretical deformation curves L_cell_(t) via the power-law rheological model. Finally, intrinsic mechanical parameters, Young’s modulus and the power-law exponent, are determined through least-squares minimization.(5)$$min\sum_{i}{\left[L_{p}\left(t_{i}\right)-L_{\mathrm{cell}\;}\left(t_{i}\right)\right]}^{2}$$

To characterize background fluid impedance for modeling prerequisites, an equivalent circuit model [18] is established, as shown in Figure 2e:(6)$$Z_{1}=Z_{1r}+jZ_{1i}=R_{e}+\left[\frac{1}{\frac{1}{R_{s}}+j\omega C_{s}}\right]$$
where j denotes the imaginary unit, ω is the angular frequency, and Z_1_ represents complex background impedance (real: Z_1r_, imaginary: Z_1i_). R_e_ characterizes electrode interfacial reactance, with the zero-phase element (ZPE) substituting conventional constant phase element (CPE) to model frequency-independent components. R_s_ and C_s_ correspond to fluid resistance and capacitance, respectively. This model accurately describes phosphate-buffered solution (PBS) electrochemical properties, providing a baseline for subsequent impedance variation analysis with biochemical additives.

For the precise simulation of experimental electrochemical impedance spectroscopy, Cao et al. [19] developed an equivalent circuit model systematically characterizing oocyte measurements at a microfluidic orifice, as shown in Figure 2f. The 12-parameter model comprises medium resistance R_s_, plasma membrane resistance R_M_, cytoplasmic capacitance C_P_, cytoplasmic resistance R_P_, perivitelline space resistance R_PVS_, zona pellucida resistances (R_1_: extra-orifice, R_2_: orifice periphery, R_3_: intra-orifice), medium resistance R_A_ in the orifice region, zona pellucida capacitance C, electrode crosstalk capacitance C_E_, and the CPE. The CPE accounts for non-ideal capacitive behavior at electrode–medium interfaces due to surface roughness.

**Figure 2 micromachines-16-00683-f002:**
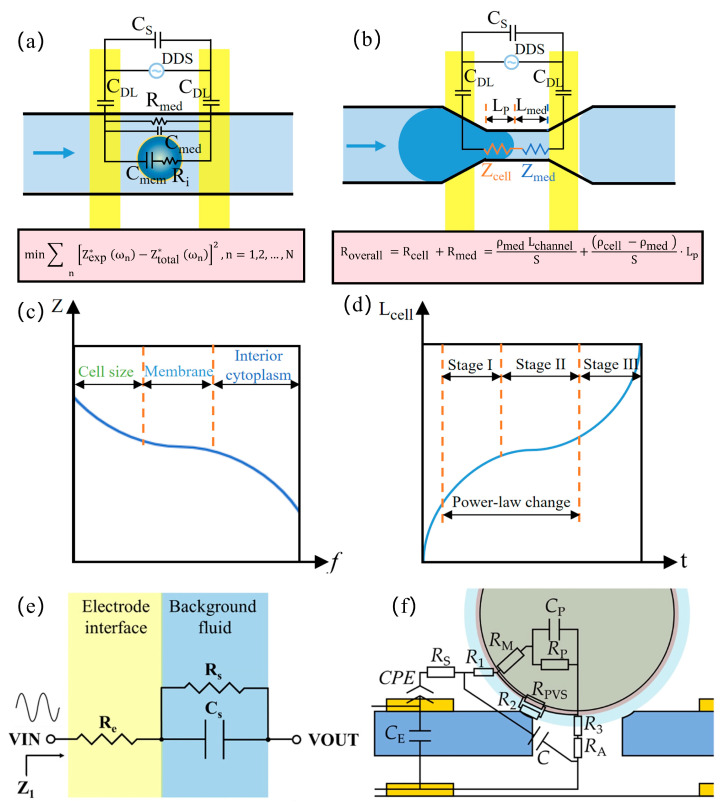
Equivalent circuit models: (**a**) an equivalent circuit diagram when measuring the electrical properties of the cell; (**b**) an equivalent circuit diagram when measuring the mechanical properties of the cell in the constricted channel; (**c**) corresponding impedance spectra showing the different dispersion dominated by R, C_sm_, and σ_i_; (**d**) an equivalent impedance–deformability mapping model of a single cell extruded through the constricted channel; (**e**) an equivalent circuit model of the background fluid (reprinted with permission from [18], published by AIP Publishing, 2025); (**f**) an equivalent circuit model at the orifice (reprinted with permission from [19], published by Elsevier, 2023).

### 2.4. Microfluidic Impedance System Design

Microfluidic impedance sensing technology achieves the high-sensitivity real-time dynamic monitoring of cells or biomolecules by integrating the precise fluid manipulation capability of microfluidic chips with the bioelectrical characteristic analysis capacity of impedance detection. The design methodology of microfluidic impedance systems is systematically introduced below through both static and dynamic configurations.

#### 2.4.1. Static System Design—Electrical Impedance Spectroscopy (EIS)

In cell culture or open microfluidic systems, where cells are maintained in static states or prolonged growth phases for extended durations, this operational paradigm is defined as static system design. Under such conditions, the detection sample is fully immersed between the detection electrodes and subjected to impedance analysis to capture temporal impedance variations over defined intervals. For static configurations, critical parameters including the volumetric dimensions of the microfluidic chip and the spatial arrangement of detection electrodes must be optimized. For example, in impedance monitoring throughout cell culture cycles, the detection electrodes are conventionally positioned on opposing sides of the cultivation chamber. During static measurements, intrinsic dielectric properties of cells such as cytoplasmic conductivity, specific membrane capacitance, Young’s modulus, cell membrane capacitance, cytoplasmic resistivity, and phase angle are quantitatively resolved.

For single cells in a flowing state, developing an effective method to achieve precise capture and in-depth analysis remains a critical challenge to be addressed. A fully electric single-cell manipulation platform was developed by Eeckhoudt et al. [20] using dual-voltage dielectrophoresis technology, which enabled the integrated control of the entire workflow including capture, retention, impedance analysis, and the release of Saccharomyces cerevisiae.

#### 2.4.2. Dynamic System Design—Impedance Flow Cytometry (IFC)

In dynamic system design, this situation refers to the high-speed movement of a biological sample within a microfluidic channel. The core principle of microfluidic impedance sensing technology lies in the fact that as targets (e.g., cells, particles) flow through the microchannel, their dielectric properties (e.g., membrane insulation, intracellular conductivity) perturb the preset AC electric field within the channel. By measuring the impedance changes (the combined effects of resistive and capacitive responses), information on the size, morphology, mechanical properties, and biochemical status of the target can be obtained. The combined advantages of this technology are threefold: real-time capability with dynamic tracking, high throughput with automation, and enhanced sensitivity and stability. This technology paradigm overcomes the static limitations of traditional impedance detection and provides a miniaturized, high-precision solution for applications such as single-cell analysis, the rapid screening of pathogens, and environmental and food monitoring. For single-cell analysis, the size of the microchannel sensing zone should be close to the size of the cell or pathogen under test for optimal accuracy; however, to avoid channel blockage, the channel size is usually slightly larger than the target size. Constricted channels are commonly used when measuring cell deformability, but such channels tend to clog. Adding xanthan gum to mixed samples can effectively reduce clogging [21]. Regarding microchannel size design, when the sensing area size matches the measured particles, the relative error of the electrical characteristic parameters measured by IFC is less than 10%. The microchannel height has the greatest impact on measurement accuracy, with channel heights exceeding the particle size increasing the relative error to 30%, while the maximum relative error caused by the electrode gap is 21.4%, slightly higher than the error caused by the channel width (20.9%) [16]. Therefore, the systematic design optimization of these parameters can further improve the measurement accuracy.

To overcome the limitations of traditional solid microchannels—such as susceptibility to microchannel occlusion and non-adjustable dimensions—Wang et al. [22] recently developed a microfluidic impedance-based flow cytometry technique based on virtual constriction microchannels. This technology constructs virtual constriction zones through the co-flow of conductive samples and insulating sheath fluids, with embedded microelectrodes positioned beneath the microchannel for impedance measurements. Compared to conventional mechanically constricted microchannels, the virtual constriction approach offers three key advantages: the effective avoidance of direct physical contact between cells and channel walls; the maintenance of high-throughput detection; and a significant reduction in the impedance sensing volume, thereby enhancing detection sensitivity. In the design of virtual constriction microchannels, the flow rate ratio of sample to sheath flow is the critical parameter. Adjusting this ratio controls the effective width of the virtual constriction zone, directly influencing both impedance detection sensitivity and throughput. The experimental results indicate that a sample-to-sheath flow ratio of 1:0.5 achieves stable fluidic focusing, with a focused stream width of 24.2 ± 0.2 μm and a theoretical detection throughput of ~1000 cells per second.

#### 2.4.3. Electrode Design

In the design of electrode configurations, two main types are used: coplanar electrodes and opposing electrodes. Coplanar electrodes are located at the bottom of the microchannel and have two modes of operation: absolute measurement (using two electrodes) and differential measurement (using three electrodes). In the differential mode, a voltage is applied through the central electrode, and the differential currents from the two side electrodes are measured to achieve a higher signal-to-noise ratio (SNR) and estimate particle velocity from the peak signal spacing. Opposing electrodes distributed at the top and bottom of the channel produce a more uniform electric field distribution, resulting in higher sensitivity than coplanar designs at the cost of increased manufacturing complexity. Additionally, liquid electrodes positioned in a transverse chamber can generate a vertical equipotential surface distribution, simplifying electric field uniformity design. The increased detection volume of liquid electrodes, however, reduces sensitivity. Each configuration presents a tradeoff between manufacturing simplicity, sensitivity enhancement, and application adaptability.

To improve the accuracy of data measurement in microfluidic impedance sensing systems, researchers have approached the problem from three aspects: electric field optimization, particle focusing, and signal compensation. Kim et al. [23] created a uniform electric field within the sensing channel by making the length of the sensing channel sufficiently long relative to the distance between the electrodes to which the potential was applied, thereby minimizing the edge effect. When focusing particles, researchers can constrain the particle trajectories through active (acoustic or dielectrophoresis) or passive (sheath flow, inertial or viscoelastic focusing) methods to ensure consistent paths through the detection area. However, this may increase the complexity of the system or dilute the samples. For this purpose, Fang et al. [24] designed a unique coplanar differential electrode device. By measuring the changes in the induced current, they successfully resolved the three-dimensional position of a single cell (with a lateral resolution of 2.1 μm and a vertical resolution of 1.2 μm), thereby avoiding the limitations of the detection flux imposed by traditional sheath flow or narrow channels. For signal compensation, calibration curves established by the correlation between electrical parameters (such as the peak width and peak height ratio) and particle position, and multi-electrode designs (such as five pairs of electrodes) or a cross wiring, are used to compensate for the signal deviation caused by position ambiguity. Zhong et al. [25] eliminated the dependence of signal sensitivity on particle height through a unique coplanar electrode configuration. In addition, customized algorithms can also be developed for special signal waveforms (such as multi-peak signals or pulse width variations) to enhance the robustness of single-cell feature extraction. These methods significantly improve the repeatability and accuracy of impedance data by reducing the errors caused by position dependence.

Additionally, static and dynamic systems can be integrated through microchannel design and the application of hydrodynamic principles. A microfluidic device integrating IFC and EIS was developed by Feng et al. [26]. Passive hydrodynamic trapping units (based on the principle of minimum flow resistance path) enabled efficient single-cell capture, while coplanar electrodes achieved the synchronous analysis of cellular electrical properties. The IFC module rapidly detected discrete impedance variations among three cancer cell lines (HeLa, HepG2, and A549), whereas the EIS module quantitatively resolved membrane capacitance and cytoplasmic conductivity. The experimental results demonstrated less than 5% deviation between IFC and EIS measurements for identical cells under flow rates of 10 nL/min to 1000 nL/min (cells were displaced from trapping sites at flow rates exceeding 10 μL/min). The EIS data maintained stability across this broad flow range, verifying IFC’s effectiveness as a high-efficiency supplement to EIS. This dual-mode strategy significantly enhanced single-cell electrophysiological characterization efficiency and reliability by synergizing the high-throughput capability of IFC with the high-resolution analytical power of EIS.

## 3. Application of Microfluidic Impedance Analysis

This chapter focuses on the diversified applications of microfluidic impedance detection technology in fields including tumor diagnostics, hematological analysis, organ-on-a-chip systems, and microbial detection. The broad application prospects spanning molecular to organ levels are comprehensively demonstrated, highlighting its significant translational value and potential from fundamental research to clinical practice. Current technical limitations are critically discussed. First, we delineate the differential characteristics between microfluidic impedance detection and three conventional detection methodologies (Table 1).

### 3.1. Tumor Detection

Tumor detection leverages microfluidic impedance sensing technology to identify, classify, and analyze tumor biomarkers, tumor cells (e.g., HeLa, HL-60, MCF-7, HepG2), or exosomes. This process focuses on elucidating tumor biology and molecular mechanisms to enable early cancer screening and precision medicine. Cancer detection leverages tumor biomarker or circulating tumor cell analysis to enable early cancer discovery and high-risk population screening; the characterization of tumor-specific cellular features clarifies molecular profiles, guiding personalized therapeutic strategies (e.g., targeted therapy, immunotherapy) to optimize efficacy while reducing adverse effects; and liquid biopsy technologies like exosome detection achieve non-invasive dynamic monitoring for evaluating treatment response and recurrence risks. Additionally, these technologies provide critical experimental platforms for drug development and mechanistic studies. Thus, cancer detection is not only pivotal for clinical diagnosis but also drives advancements in personalized oncology and cancer prevention.

In the static design of microfluidic impedance detection systems, addressing the challenge of detecting tumors at very early or asymptomatic stages, Hu et al. [33] developed a porous 3D graphene aerogel biochip and combined it with EIS technology to achieve the ultra-sensitive detection of alpha-fetoprotein (LOD = 7.9 pg/mL) and carcinoembryonic antigen (LOD = 6.2 pg/mL), and the detection limit of tumor exosomes was as low as 10 particles/mL, surpassing the performance of commercial immunoassay kits. When performing dynamic detection on cells, to improve the accuracy of measurement data, it is generally necessary to focus the cells. Zhu et al. [34] used a virtual sensing zone using polyethylene glycol (PEG) as a sheath flow, which increased the detection sensitivity and signal-to-noise ratio by 7.92 times and 1.42 times, respectively, significantly better than the traditional contraction channels. Traditional microfluidic channels are prone to clogging in the contracted part because their size cannot be changed. To address channel clogging, Li et al. [21] developed an anti-clogging microfluidic impedance cytometer with xanthan gum-based cell pretreatment, achieving 240 cells/s throughput and leveraging the triphasic mechanical dynamics (creep, friction, and relaxation) of long constriction channels to analyze the complex mechanical properties of single cells, thereby enhancing the measurement accuracy of mechanical characteristic parameters, as shown in Figure 3a. Additionally, Luan et al. [35] introduced a parallel physical fitting solver to analyze specific membrane capacitance and cytoplasmic conductivity at a high speed of 1613 cells/s, which is 27,000 times faster than traditional methods, as shown in Figure 3b.

However, when analyzing cell parameters and then classifying cells, a single mechanical or electrical characteristic parameter is not sufficient to classify cells more accurately. To solve this problem, by simultaneously analyzing the five-dimensional intrinsic parameters of single cells, including Young’s modulus, fluidity, cell radius, cytoplasmic conductivity, and specific membrane capacitance, Feng et al. [36] achieved a high-precision classification rate of 93.4% for three cancer cell subtypes (HepG2/MCF-7/MDA8) and 95.1% for the cytoskeleton perturbation model (fixed/treated with cytochalasin B), as shown in Figure 3c. In pursuit of higher classification performance, Tang et al. [37] developed an asymmetric serpentine microchannel chip and combined various dielectric parameters (cell diameter, impedance amplitude, impedance phase shift and electrical opacity) with machine learning classification models to identify cell types, achieving 96.2–99.6% accuracy in discriminating human leukocytes from four tumor cell lines (A549, MCF7, H226, and H460). This work demonstrated the data-mining potential of machine learning for uncovering subtle cellular signatures hidden in multi-parameter dielectric datasets. To enable the rapid detection of tumor cells in human blood for improving early cancer diagnosis rates or treatment efficacy assessment, Zhu et al. [38] developed an automated liquid biopsy device integrating inertial microfluidics and impedance sensing. Combined with a deep learning algorithm, this system achieved the label-free detection of circulating tumor cells in peripheral blood within 15 min. When analyzing a cell population, measuring the overall viability of the cells is a difficult problem, Xie et al. [39] designed a microfluidic impedance system to quantify the toxicity of pollutants to three-dimensional human liver cancer cell clusters (HepG2) through the cell clustering index, and proved that it can be used as a measurement indicator of overall cell viability. In addition, mass spectrometry analysis technology based on microfluidic chips has further improved the precision analysis of single cell internal substances. For mass spectrometry analysis technology, it is usually necessary to purify and desalt the samples. Zhu et al. [40] developed a one-step impedance flow cytometry protocol integrating cell sorting and desalting (>99% efficiency), resolving pre-MS purification and salt interference simultaneously, as shown in Figure 3d.

**Figure 3 micromachines-16-00683-f003:**
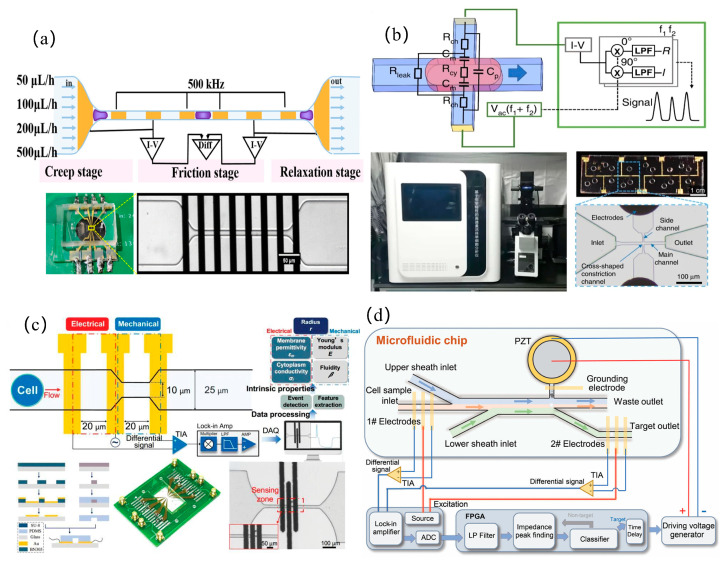
Tumor detection: (**a**) microfluidic impedance flow cytometer featuring xanthan gum pretreatment protocol to mitigate microchannel occlusion in constriction-based cellular interrogation systems (reprinted with permission from [21], published by American Chemical Society, 2024); (**b**) ultrahigh-throughput single-cell biophysical analyzer employing parallel physics-informed solver architecture for concurrent quantification of specific membrane capacitance and cytoplasmic conductivity (reprinted with permission from [35], published by Springer Nature, 2023); (**c**) multimodal microfluidic system with integrated electromechanical sensing modules for concurrent mechanoelectrical phenotyping of cellular intrinsic properties (reprinted with permission from [36], published by Wiley-VCH Verlag, 2023); (**d**) integrated microfluidic impedance cytometer implementing label-free operation mode with single-step analytical process combining cellular characterization and on-chip desalination (reprinted with permission from [40], published by Wiley-VCH Verlag, 2024).

### 3.2. Blood Detection

Blood detection employs microfluidic impedance sensing technology to identify, classify, and analyze blood components (e.g., red/white blood cells, proteins, biomarkers), enabling the assessment of individual health status, disease diagnosis, or therapeutic efficacy monitoring. Its clinical significance lies in the early detection of conditions such as anemia, infection, diabetes, and hepatic/renal dysfunction while tracking chronic disease progression or treatment outcomes to guide therapeutic adjustments; the dynamic monitoring of blood parameters also identifies latent health risks, supporting disease prevention and health management. As an indispensable core tool in clinical diagnostics and population screening, it integrates precision, non-invasiveness, and real-time capabilities to advance personalized healthcare.

Innovations in single-cell and blood parameter detection have driven progressive breakthroughs across research teams. Caselli et al. [41] pioneered neural network-based raw impedance data parsing to extract intrinsic dielectric properties (cell radius, membrane capacitance, and cytoplasmic permittivity/conductivity) at millisecond resolution, resolving signal overlap distortions in high-throughput single-cell analysis while capturing hidden single-cell signals, as shown in Figure 4a. Advancing multi-parameter blood analysis, Zhbanov et al. [42] developed a flow-mode microfluidic dielectric spectroscopy sensor, combining effective medium theory to simultaneously quantify six parameters (RBC count, hemoglobin, etc.) with <3.5% clinical error, suppressing RBC aggregation via flow impedance and introducing an orientation factor to quantify RBC alignment.

In the pursuit of optimizing blood cell detection performance, research teams have propelled progress by breaking through the limitations of traditional methods through multidimensional innovations. Huang et al. [43], through dielectrophoretic (DEP) force manipulation and a uniquely engineered structural design, achieved high mixing efficiency, enabling the real-time measurement of dynamic permeability changes in single red blood cells within 0.19 s. Zhong et al. [25] eliminated the dependence of sensitivity on cell height via a novel coplanar electrode configuration, achieving high-throughput detection at 1000 cells/s. The position-insensitive characteristics of this design not only enabled the precise classification of three leukocyte subtypes, but also simultaneously acquired red blood cell indices while avoiding traditional hemolysis steps, as shown in Figure 4b. To further enhance classification accuracy, Wang et al. [22] constructed a dynamic detection zone via sheath flow crossover and achieved 99.8% classification accuracy for three leukemia cell types (K562, Jurkat, and HL-60) and ≥99.2% recognition precision for four leukocyte subtypes (NEU/EOS/MON/LYM) using a recurrent neural network.

In the development of integrated microfluidic devices, research teams are progressively advancing the practicality and scenario adaptability of detection technologies through modular design and functional integration. Oshabaheebwa et al. [44] developed a fully miniaturized, portable, and wash-free microfluidic impedance detection platform capable of generating an erythrocyte occlusion index within 15 min, enabling the rapid assessment of red blood cell health, functional status, and therapeutic efficacy, as shown in Figure 4c. Fu et al. [45] developed an origami-inspired electrochemical microfluidic paper-based device that integrates plasma separation, buffer absorption, and three-channel EIS detection. This system achieved the simultaneous detection of three cardiac protein markers (LODs: 4.6/1.2/146 pg/mL) in fingerprick whole blood, delivering rapid analysis within 46 min—a performance comparable to commercial ELISA kits, as shown in Figure 4d. To address the challenge of stroke subtyping, Sayad et al. [46] designed a magneto-impedance biosensor combined with a microfluidic chip for glial fibrillary acidic protein (GFAP) extraction from plasma. By leveraging Dynabeads magnetic labels, the system achieved a detection sensitivity of 1.0 ng/mL for GFAP in acute stroke patient blood samples, fulfilling the clinical need for subtype differentiation. To address the issues of time-consuming diagnostics and frequent false positives in urinary tract infection (UTI) detection, Petchakup et al. [47] developed a microfluidic inertial impedance sensing platform. This system enables the label-free sorting of neutrophils directly from urine samples in 5 min, followed by impedance spectroscopy analysis, thus facilitating rapid, culture-free urine screening for infection biomarkers.

**Figure 4 micromachines-16-00683-f004:**
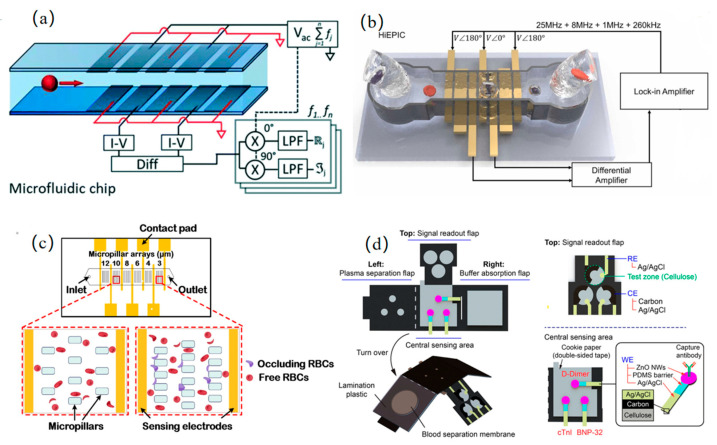
Blood detection: (**a**) neural network-based decoding of raw impedance data streams enabling extraction of single-cell signatures hidden within overlapping cellular measurements (reprinted with permission from [41], published by Royal Society of Chemistry, 2022); (**b**) microfluidic chip with unique coplanar electrode configuration achieving concentration-independent sensitivity through elimination of high-concentration dependency (reprinted with permission from [25], published by Elsevier, 2022); (**c**) portable, wash-free microfluidic platform for multiplex assessment of erythrocyte health, functional characterization and therapeutic efficacy monitoring within 15 min (reprinted with permission from [44], published by Elsevier, 2024); (**d**) origami-inspired electrochemical microfluidic paper-based device for simultaneous quantitation of three cardiac protein biomarkers in fingerprick whole blood without pretreatment (reprinted with permission from [45], published by American Chemical Society, 2023).

### 3.3. Organ on a Chip

Organ on a chip (OOC) is a biomimetic microfluidic platform that replicates the physiological functions and microenvironment of human organs, providing an efficient and cost-effective experimental tool for drug screening, toxicology studies, and disease modeling. Traditional in vitro models fail to simulate the complex structures and dynamic interactions of organs, while animal studies face ethical constraints and interspecies differences. To bridge these gaps, OOC technology integrates microfluidics to precisely regulate fluid flow, mechanical forces, and biochemical signals and incorporates impedance sensing to monitor cellular dynamics and functional responses in real time, enabling a paradigm-changing approach to organ-level physiological and pathological studies. In recent years, OOC systems that integrate microfluidic and impedance analysis technologies have been widely used in vascular, brain, liver, kidney, and intestinal models, establishing high-fidelity in vitro platforms for precision medicine and personalized drug development. The convergence of microfluidic and impedance technologies highlights their transformative potential to advance biomedical research and therapeutic innovation.

The core design of OOC systems lies in the deep integration of impedance sensing and microenvironmental regulation. Such architecture provides a high-fidelity in vitro platform for advancing precision medicine and personalized drug development by recapitulating tissue-specific physiological and pathological dynamics. As the body’s central transport network, vascularization on a chip further enables the interconnection of multi-organ chips, thereby reconstructing a more physiologically complete in vitro system. Na et al. [48] proposed a hemodynamic similarity principle that enables the rapid derivation of the input impedance for in vitro microgravitational systems derived from in vivo arterial systems, thereby accurately replicating the desired endothelial hemodynamic microenvironment. To address the challenge of the dynamic monitoring of tumor angiogenesis, Huang et al. [49] developed a 3D biomimetic model based on matrix gel microchannels. By integrating microelectrodes to quantify impedance changes during vascular extension, they demonstrated a linear relationship between the vascular growth rate and vascular endothelial growth factor concentration and revealed the regulatory role of cancer cell count in angiogenesis, as shown in Figure 5a.

The brain, serving as the central control hub for the entire body’s functions, plays a critically important role in regulating human physiological activities. Liang et al. [50] developed a bioinspired brain microenvironment chip by integrating interstitial flow regulation and real-time neural circuit monitoring. This system deciphered the dynamic characteristics of synaptic connection/disconnection processes and established a highly biomimetic model for studying the mechanisms of neurodegenerative diseases, as shown in Figure 5b. In addition, Ceccarelli et al. [51] designed a blood–brain barrier (BBB) chip integrated with thin-film electrodes, utilizing EIS to track the integrity and maturation of the BBB in real time. This innovation provides a highly efficient in vitro platform for investigating drug permeation mechanisms in neurological disorders. As the central metabolic hub for both catabolism and anabolism, the liver plays a pivotal role in detoxification and biosynthesis, and studying its function is essential for elucidating disease mechanisms such as metabolic disorders, fibrosis, and hepatocellular carcinoma. Dhwaj et al. [52] developed a liver-on-a-chip system powered by a low-power impedance micropump to enable continuous nutrient perfusion, mimicking cardiac pulsation-driven medium circulation for the real-time monitoring of healthy cell growth over 14 days. The compact design (compatible with standard Petri dishes) eliminates contamination risks associated with conventional perfusion systems.

As the central filtration hub of the blood, the kidneys are pivotal for maintaining systemic homeostasis. The development of in vitro renal models is critical for advancing therapies for uremia and other renal pathologies. Liang et al. [53] developed a microfluidic impedance sensing system to dynamically track the formation and drug response of renal tubular epithelial barriers. They demonstrated that basement membrane matrix (ABM)-supported barriers exhibited significantly superior monolayer stability and growth rates compared to Transwell membrane scaffolds. The platform validated its high sensitivity to biochemical stimuli through calcium switch assays and employed equivalent circuit modeling to correlate barrier impedance with cellular density and junctional tightness, as shown in Figure 5c. As the terminal site for nutrient absorption, intestinal dysfunction has garnered significant attention due to the growing prominence of modern food safety issues (e.g., chemical additives, microbial contamination). Fernandes et al. [54] developed a multi-channel microfluidic system that monitored the growth and polarization processes of human epithelial cells by conducting continuous impedance measurements over five consecutive days, enabling the real-time tracking of epithelial barrier integrity, as shown in Figure 5d. To address the limitations of traditional models in simulating microenvironments with varying oxygen concentrations, Khalid et al. [55] proposed a method using inkjet-printed embedded dissolved oxygen (DO) and reactive oxygen species (ROS) electrochemical sensors. This approach was implemented on a custom-developed bilayer gut-on-a-chip platform to monitor both developmental phases (initially normoxic) and induced hypoxia over 6 days.

**Figure 5 micromachines-16-00683-f005:**
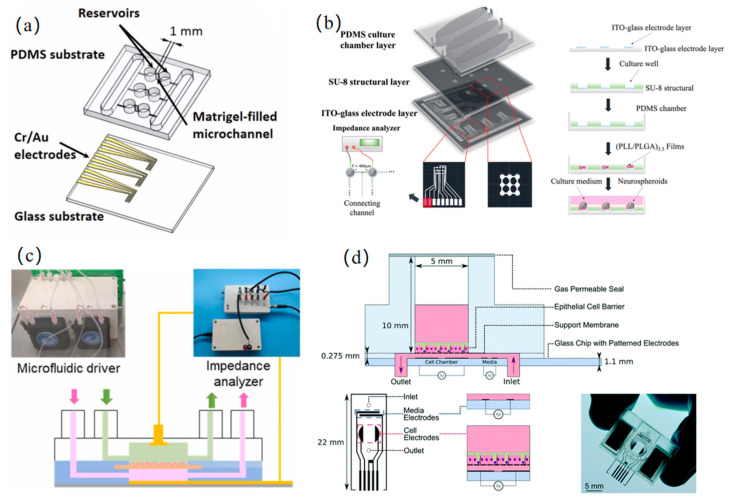
Organ on a chip: (**a**) a 3D biomimetic model based on matrix gel microchannels for the dynamic monitoring of tumor angiogenesis (reprinted with permission from [49], published by Elsevier, 2022); (**b**) a brain-on-chip biomimetic microenvironment with interstitial flow regulation and neural network analysis for the dynamic characterization of synaptic connection/disconnection kinetics (reprinted with permission from [50], published by the Royal Society of Chemistry, 2024); (**c**) a microfluidic platform for the spatiotemporal tracking of renal tubular epithelial barrier formation and pharmacodynamic evaluation (reprinted with permission from [53], published by Elsevier, 2023); (**d**) a multi-channel microfluidic array for the real-time assessment of epithelial barrier integrity (reprinted with permission from [54], published by the Royal Society of Chemistry, 2022).

### 3.4. Microbial Detection

Microbial detection is defined as the process of qualitative and quantitative analysis for microorganisms (viruses, bacteria, fungi, and algae) using microfluidic impedance sensing technology. Its principal value lies in providing critical data support for disease control, food safety, and environmental protection through the precise identification of microbial characteristics. In healthcare applications, pathogens can be rapidly localized using this technique, enabling the rational guidance of antibiotic usage and the containment of infectious disease transmission. Within the food industry, microbial contamination-induced foodborne illnesses are effectively prevented, thereby ensuring food safety. For environmental monitoring, specific marine algae species are monitored to provide early warnings of ecological imbalance and environmental pollution risks. Furthermore, in biopharmaceutical and agricultural sectors, the dynamic monitoring of microbial variations is implemented through this technology, facilitating process optimization and risk management. These multidimensional capabilities establish it as a pivotal tool for safeguarding human health and achieving sustainable development.

In microbial detection, the detectability of microbial throughput or concentration is recognized as a pivotal performance metric. Regarding throughput metrics, advancements have been systematically pursued. Chen et al. [56] constructed a 3D detection zone with cylindrical electrodes and microchannels, achieving a throughput of 1800 Haematococcus pluvialis cells per second, as shown in Figure 6a. For concentration range optimization, expanding the detectable dynamic range becomes imperative to enhance analytical versatility. Chen et al. [57] developed a microfluidic system with an ultra-wide detection range (615–2.7 × 10^8^ cells/mL), simultaneously differentiating bacterial viability and Gram types (LOD = 10^2^ cells/mL). Microfluidic impedance sensing technology enables not only single-bacterium detection but also concurrent multi-pathogen discrimination. Muhsin et al. [58] optimized microfluidic layouts to concurrently detect Salmonella, Legionella, and E.coliO157:H7 (LOD = 3 cells/mL, 30–40 min detection, no pre-enrichment). For rapid bacterial antibiotic resistance testing, Chen et al. [59] created a single-bacterium impedance detection platform. Antibiotic susceptibility tests were completed within 32 min (30 min antibiotic exposure added to 2 min impedance analysis), significantly accelerating detection cycles, as shown in Figure 6b. However, the discrimination between common bacteria and drug-resistant bacteria is recognized as a significant technical challenge. Tang et al. [60] developed parallel impedance cytometers with dual microchannels, where opposing impedance signals in synchronized time sequences enabled high-precision identification. Bacterial classification is recognized as a technically demanding process due to submicron dimensions, presenting greater technical challenges compared to conventional eukaryotic cell analysis. Convolutional neural networks were employed by Zhang et al. [61] to analyze impedance flow cytometry data, achieving >96% accuracy in bacillus/coccus/vibrio classification and >95% precision in Escherichia coli vs. Salmonella enteritidis discrimination, as shown in Figure 6c. However, current methodologies that constrain idealized sample validation, without addressing the complexity of real-sample matrices, may pose critical technical barriers for clinical translation. Piedimonte et al. [62] created a differential impedance system with gold microelectrodes and nanoadsorbed antibodies, achieving 88 pg/mL anti-dengue antibody sensitivity (superior to commercial ELISA). The reconfigurable probe design further enabled the rapid screening of other pathogens and diseases, as shown in Figure 6d. Almalaysha et al. [63] demonstrated 1–2 cells/mL Salmonella typhimurium detection in raw chicken, validating real-sample reliability. An impedance spectroscopy-based electronic tongue was engineered by Coatrini-Soares et al. [64], where optimized sensing units fabricated through layer-by-layer thin films were combined with machine learning algorithms employing decision tree models to process capacitive data, achieving the sensitive detection of Staphylococcus aureus in milk (LOD = 2.01 CFU/mL). Dowdell et al. [65] integrated online flow cytometry into water ozonation systems, successfully predicting microbial anomalies through the 40-day continuous monitoring of total/intact cell counts.

Beyond the detection targets, microfluidic impedance sensing technology enables the high-sensitivity and non-invasive analysis of other microscopic biological samples (e.g., stem cells, oocytes, sweat, and microplastics). Stem cell and oocyte detection is utilized to assess cellular viability and quality in regenerative medicine or assisted reproduction, thereby optimizing therapeutic outcomes. Cam et al. [66] quantified the therapeutic window of bone marrow mesenchymal stem cell exosomes through real-time impedance analysis, with their repair efficacy and functional recovery capabilities for ischemic acute kidney injury structures being precisely evaluated. Due to long-standing limitations in oocyte quality assessment caused by the subjectivity of zona pellucida hardening detection, Cao et al. [19] developed a microfluidic cell capture technique where impedance variations were correlated with zona pellucida aspiration lengths. This approach allowed the calculation of the zona pellucida’s Young’s modulus and the quantitative characterization of its aspiration process at the opening.

Sweat analysis not only achieves the dynamic monitoring of electrolytes, metabolites, and inflammatory factors but also provides real-time data for sports medicine, dermatological disorders, and chronic disease management. In wearable device research, human sweat analysis is predominantly conducted. Wang et al. [67] engineered a self-powered multifunctional microfluidic sweat analysis system where spontaneous sweat transport was driven by wettability gradient design. By integrating impedance–colorimetric dual-mode sensing, the simultaneous detection of sweat flow rate, electrolyte concentration, glucose, and pH was accomplished with the response time reduced to 100 ms. To enhance wearable system practicality, Zahed et al. [68] constructed a wireless monitoring patch system using flexible materials, integrating microfluidic channels with multimodal sensing. The reduced graphene oxide glucose sensor exhibited an enhanced sensitivity of 19.97 μA/mM/cm^2^ (2.3-fold improvement versus non-microfluidic designs) under microfluidic assistance, achieving the synchronous dynamic monitoring of sweat glucose and ECG signals during exercise while validating stability and bend resistance in complex scenarios. With increasing demands for marine pollution monitoring, microplastic identification in seawater has emerged as a critical challenge. Butement et al. [69] implemented an impedance-based cytometer combined with machine learning algorithms, enabling the direct recognition and counting of microplastic particles within phytoplankton mixtures (1.5–10 μm) in seawater-mimicking media. For practical sample detection, Silva et al. [70] developed an impedance-based microfluidic electronic tongue. Multi-sensing units coated with nanostructured thin films were integrated, and chemometric methods including principal component analysis and partial least squares-discriminant analysis were employed. The accurate discrimination of authentic coconut water samples (classification accuracy >90%) was achieved, along with the high-precision prediction of physicochemical parameters such as soluble solid content and total titratable acidity. Xue et al. [71] developed a copper nanoparticle-enhanced microfluidic flow injection analysis system. Electrochemical impedance spectroscopy was employed to reveal a 20-fold lower charge transfer resistance at copper nanoparticle-modified multi-walled carbon nanotube screen-printed carbon electrodes versus standard counterparts. Microfluidic parameters were optimized and this modified electrode was utilized to achieve highly sensitive dopamine (DA) detection (LOD = 0.33 nM). Across three authentic sample matrices—artificial cerebrospinal fluid, fetal bovine serum, and human urine—recovery rates of 96.5% to 103.8% were demonstrated, proving its practical utility. These technological advancements not only expand conventional detection dimensions but also demonstrate precision and operational efficiency, significantly accelerating the translation of fundamental research into clinical applications.

Notwithstanding the multi-domain application potential demonstrated by microfluidic impedance sensing technology, the following critical bottlenecks persist: Practical application validations remain insufficient for most current technologies. High-sensitivity and high-classification-accuracy performance metrics are predominantly validated under laboratory-standardized conditions using simplified models (e.g., single-cell lines or purified samples). However, in complex biological fluids (e.g., whole blood, interstitial fluid, or clinical samples with high background interference), performance robustness and reliability are compromised by non-target component interference, matrix effects, and dynamic physiological environmental factors. Although machine learning significantly enhances classification efficiency, models exhibit pronounced dataset-specific dependency and limited generalization capability for unknown sample types. The opacity of black-box models hinders biologically interpretable explanations, undermining clinical decision-making credibility. Practical implementation is hindered by complex structural dependencies (e.g., virtual contraction channels, or sheath flow control), resulting in difficulties in device miniaturization, elevated manufacturing costs, and operational complexity. These constraints conflict with the portability and user-friendliness requirements of POCT; Despite advancements in simultaneous multidimensional physical/electrical characterization, accuracy is affected by parameter coupling effects. The absence of unified data fusion standards and cross-platform comparability further impedes large-scale technological deployment. These challenges collectively emphasize the necessity for future research to prioritize clinical translation, technical simplification, multimodal data standardization, and system stability optimization.

## 4. Conclusions and Outlook

In summary, this review systematically examines recent advancements in the integration of microfluidic and impedance sensing technologies, along with their technological evolution in biosensing applications. Starting from the fundamental principles of microfluidic technology and impedance spectrum analysis, the detection mechanisms based on fluidic characteristics and impedance analysis were introduced, with particular emphasis on the construction logic of cellular impedance equivalent circuit models. Detection systems were classified into static and dynamic categories according to cellular motion states, with their design principles and critical factors for enhancing detection accuracy being elucidated, respectively. Finally, the innovative applications were systematically categorized into four domains: tumor detection, blood detection, OOC, and microbial detection based on biological target dimensions.

Despite critical progress achieved through microstructural and electrode design optimizations, several challenges remain (Figure 7).

(1)Signal Noise and Interference Suppression: Signal noise represents a critical challenge affecting data accuracy and system reliability during microfluidic impedance measurements. Due to the compact size and high sensitivity of microfluidic devices, susceptibility to external electromagnetic interference exists, particularly in open measurement environments. Furthermore, device-intrinsic parasitic capacitance, poor contacts, and electrode contamination introduce additional noise, degrading signal quality. Differential measurement techniques are typically employed to effectively suppress common-mode noise through differential signal extraction for automatic noise balancing. Shielding materials and electromagnetic isolation methods, such as copper foil tapes, are utilized to minimize external interference. Signal processing approaches including filtering algorithms and adaptive noise cancelation algorithms have been implemented for post-processing raw signals to enhance signal quality and stability. Although numerous studies attempt to model and analyze impedance data using statistical methods and machine learning techniques, limitations persist in generalization capabilities and practical applicability due to insufficient data volume and sample imbalance. Particularly, machine learning algorithms require extensive annotated datasets for training, which are often unavailable in real-world scenarios.(2)Device Miniaturization and Standardization: While microfluidic technology inherently offers miniaturization advantages, achieving further size reduction becomes complex when integrating multi-channel, multi-parametric detection systems. Conventional microfluidic systems typically rely on single electrode arrays for impedance measurements, whereas practical applications necessitate integrated multiple electrodes and intricate microchannel designs to accommodate diverse detection needs. The miniaturization of impedance analyzers remains challenging, impeding portable system development. Additionally, standardization deficiencies exist since most microfluidic chips are laboratory-customized via photolithography without uniform specifications for electrode dimensions/materials. This variability hinders standardized performance evaluation across methodologies.(3)Biological Sample Pre-processing: Microfluidic impedance technology has significant applications in biomedical detection, particularly for the rapid analysis of biological samples like blood and urine. However, the inherent heterogeneity of biological samples, including cell density, protein concentration, and ionic strength, directly affects impedance measurements. Sample pre-processing is crucial in practice. For instance, whole blood contains abundant cellular components and solutes; direct measurement may cause signal confusion due to conductivity differences between cell membranes and solutions. Accuracy enhancement often requires pre-processing steps such as dilution, centrifugation, or filtration. Consequently, designing integrated pre-processing zones (e.g., filtration/separation units) within microfluidic chips is essential.

Based on our discussion, the future development of microfluidic impedance sensing technology will focus on two core objectives: “practicality optimization” and “clinical adaptability enhancement”.

(1)Standardization and manufacturing simplification must be prioritized: Unified chip design specifications (the electrode layout and channel dimensions) and detection protocols should be established to facilitate laboratory-to-industry translation. Cost-effective microfabrication techniques (e.g., lithography injection molding scale production) and novel electrode materials (flexible conductive polymers, carbon-based composites) require development to reduce precious metal dependency while improving batch consistency and mechanical stability.(2)Anti-interference capability and portability require targeted improvements: Integrated pretreatment modules (e.g., embedded filtration membranes) should be optimized to minimize matrix interference in complex samples (whole blood and saliva). Miniaturized signal processing units combined with low-power circuits must evolve toward handheld/wearable formats to meet POCT demands. Bioresorbable material innovations enabling implantable monitoring devices demand breakthroughs in long-term signal drift correction and biosafety evaluation through fundamental studies on material-biological interface dynamics.(3)Artificial intelligence (AI) systems and data processing strategies: AI and machine learning (ML) are increasingly being used in the data analysis and processing of microfluidic impedance chip systems [72,73,74]: Convolutional Neural Networks (CNNs) automatically extract spatial features (e.g., cellular morphology, impedance distribution), where local patterns are captured through convolutional layers while dimensionality reduction and noise resistance are achieved via pooling layers. Raw images/signals are directly processed to identify cell-type differences, and multi-source parameters (temperature/pH) are fused to enable real-time sensor drift correction. Recurrent neural network variants (GRU/LSTM) model temporal dynamics (e.g., impedance changes during cell division, biomarker fluctuations in sweat) through gating mechanisms, with long-term dependencies captured to predict physiological trends. These are combined with CNNs to form hybrid architectures (CNN-GRU) for spatiotemporal feature co-analysis. These methods further enhance the ability of impedance signals to predict cell behavior and response, accurately predict future dynamic changes in cells through real-time and historical data and greatly enhance the diagnostic performance of microfluidic impedance analysis technology. In addition, AI algorithms can also be used to control microfluidic impedance systems, form intelligent feedback mechanisms, adjust experimental conditions (such as the flow rate, measurement frequency, etc.) in real time, perform high-precision measurements of biological samples of different sizes and types, and achieve closed-loop control and adaptive adjustment. Furthermore, the AI method of integrating impedance analysis with multimodal data fusion such as optical imaging, biosensors, and Raman spectroscopy analysis is expected to further improve the sensitivity and specificity of microfluidic chips in diagnosis and detection.(4)AI-driven impedance microfluidic chip design: With the development of AI intelligent design microfluidic chips, as well as artificial intelligence expert models based on large language models such as ChatGPT [73,75], the development of impedance microfluidic technology chip design AI and expert models will become the key to the next generation of impedance microfluidic technology. This will greatly lower the threshold for the application of impedance microfluidic chips and make the technology accessible to a wider range of biological research fields.(5)Multimodal integration: Synergistic combinations with acoustic, magnetoelastic, and Raman spectroscopic techniques should create complementary multidimensional sensing. Self-powered systems, surface functionalization, self-cleaning microstructures, and modular replaceable electrodes require integration to extend chip lifespan and reduce maintenance costs.

These pragmatic improvements will propel microfluidic impedance sensing technology from laboratories into clinical, household, and industrial settings, ultimately establishing it as a practical tool for precision medicine and health monitoring with global health impacts. This review adheres to the PRISMA (Preferred Reporting Items for Systematic Reviews and Meta-Analyses) standards (Appendix A).

## Figures and Tables

**Figure 1 micromachines-16-00683-f001:**
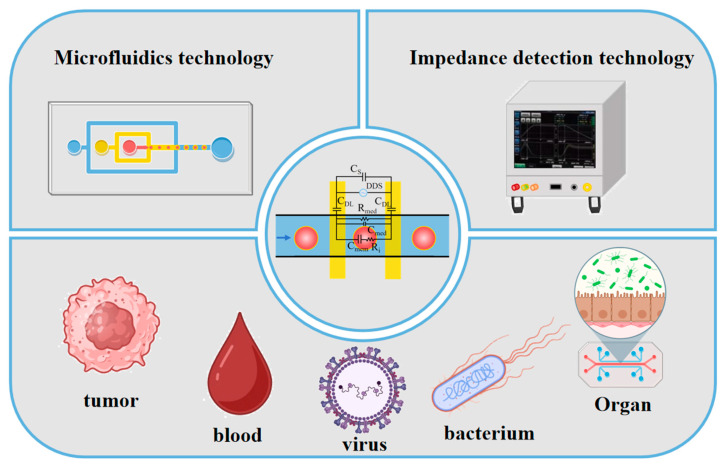
Innovative applications of microfluidic impedance detection technology in various fields.

**Figure 6 micromachines-16-00683-f006:**
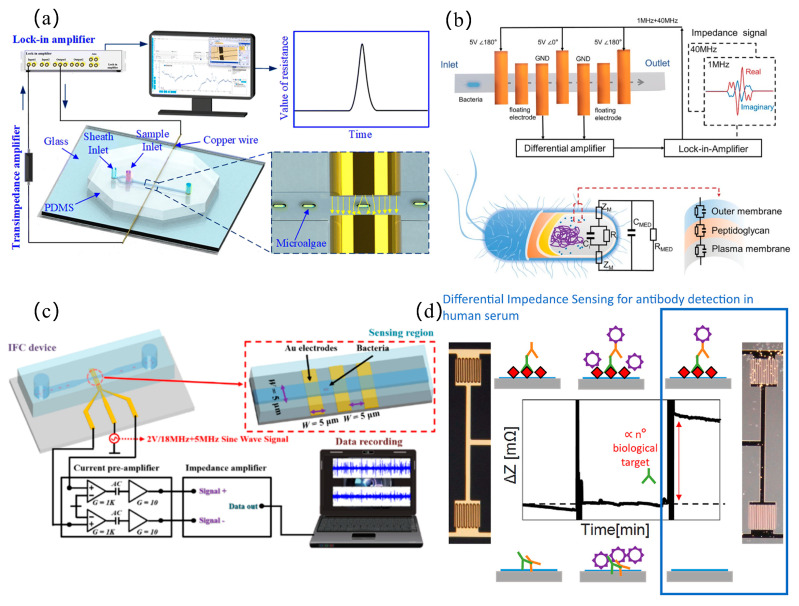
Microbial detection: (**a**) 3D-architected microfluidic impedance cytometer with dual cylindrical electrodes and embedded microchannels for enhanced cellular interrogation (reprinted with permission from [56], published by Royal Society of Chemistry, 2024); (**b**) single-bacterium-resolution impedance cytometry platform enabling rapid antimicrobial susceptibility testing via electrophysiological profiling (reprinted with permission from [59], published by Wiley-VCH Verlag, 2024); (**c**) microfluidic device integrated with convolutional neural network-based deep learning framework for automated interpretation of impedance flow cytometry data to achieve precision bacterial classification (reprinted with permission from [61], published by American Chemical Society, 2024); (**d**) differential impedance biosensing system implementing phase-sensitive signal demodulation for label-free detection of anti-dengue viral antibodies in low-concentration sera (reprinted with permission from [62], published by Elsevier, 2022).

**Figure 7 micromachines-16-00683-f007:**
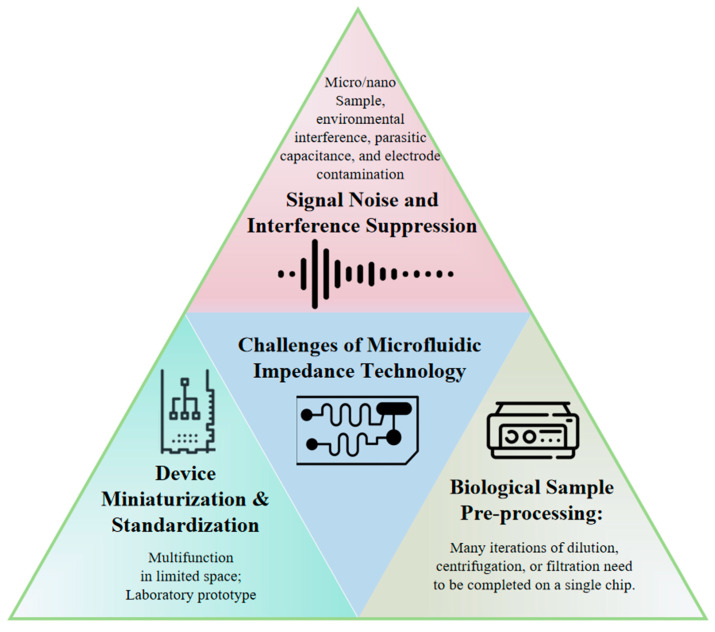
Conceptual map of prevailing challenges in microfluidic impedance detection technology.

**Table 1 micromachines-16-00683-t001:** Comparison of microfluidic impedance detection technology with three other detection methods.

Category	Microfluidic Impedance Detection	Fluorescence Detection [27,28]	PCR [29,30]	ELISA [31,32]
Principle	Label-free electrical detection based on dielectric property differences in cells/particles	Fluorescent dye labeling with optical signal acquisition	Nucleic acid amplification and fluorescence quantification	Antigen–antibody binding with enzymatic chromogenic reaction
Advantages	Label-free operation, real-time dynamic monitoring, miniaturized equipment, low sample consumption, simplified workflow	High sensitivity (single-molecule level), multi-channel parallel detection, mature technology	Ultra-high sensitivity (aM–fM), exceptional specificity, quantitative capability	High specificity, high throughput, standardized protocols, commercial maturity
Limitations	Susceptible to environmental interference, limited multi-target detection capability, complex chip design requirements	Label-dependent (increased cost/time), photobleaching issues, expensive instrumentation	Thermal cycler dependency, complex instrumentation, contamination risks, non-nucleic acid targets undetectable	Antibody quality dependency (potential cross-reactivity), lengthy procedures, limited small-molecule detection
Detection time	Minutes (real time)	Minutes/hours	1–3 h	3–24 h
Sensitivity	pg/mL level	fg/mL level	aM–fM level	pg/mL level
Throughput	Medium–high (chip-dependent)	Medium	Medium	High
Cost	Low equipment cost, high chip development cost	High reagent cost (labels), high equipment cost	High equipment/reagent costs	Moderate equipment/reagent costs
Applications	Point-of-care testing, cellular analysis	Single-molecule detection, imaging analysis	Pathogen detection, gene expression	Protein biomarkers, clinical diagnostics

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
