# Peer review of "Recent Advances in Microfluidic Impedance Detection: Principle, Design and Applications"

_micromachines, 2025, doi:10.3390/mi16060683_

Round 1

Reviewer 1 Report

Comments and Suggestions for Authors

Herein, the article provides a comprehensive and detailed review of microfluidic impedance detection technology. The article concludes with a forward-looking perspective, highlighting future directions such as AI-driven algorithms and flexible electronics integration, emphasizing a clear roadmap for future research and development in the field. However, there are areas where the manuscript could be improved to make it more impactful.

Comments:

  1. The review could benefit from simplified explanations or visual aids (e.g., diagrams, flowcharts) to make the technical content more accessible to a broader audience. While the depth of detail is impressive, the use of complex jargon and mathematical formulations may disengage readers who are not experts in the field.
  2. A comparative analysis of microfluidic impedance detection with other biosensing technologies (e.g., fluorescence-based assays, PCR, ELISA) would enhance the review. A side-by-side comparison of advantages, limitations, and performance metrics would help readers better understand the unique value proposition of this technology. The authors can consider adding some tables for the comparative analysis.

  3. The discussion on AI and machine learning integration is limited. While the article mentions AI-driven algorithms as a future direction, it does not provide detailed insights into how these technologies are currently being used to enhance data analysis and interpretation in microfluidic impedance systems.

  4. The review could include more real-world validation and case studies to demonstrate the practical impact of microfluidic impedance detection. Examples of its use in real-world settings would provide valuable insights into its scalability and challenges.

  5. The article briefly mentions technical bottlenecks (e.g., signal noise suppression, device miniaturization) but does not delve deeply into the practical challenges of implementing the technology in real-world settings. A more detailed exploration of these challenges would provide a more comprehensive understanding of the barriers to widespread adoption of this technology.
Comments on the Quality of English Language

The English in the article is clear and well-written, but some sections are overly technical and complex. Simplification of the language, especially in parts with heavy jargon or math, would make it more accessible to a wider audience, including non-experts.

Author Response

    1.    The review could benefit from simplified explanations or visual aids (e.g., diagrams, flowcharts) to make the technical content more accessible to a broader audience. While the depth of detail is impressive, the use of complex jargon and mathematical formulations may disengage readers who are not experts in the field.

    Reply 1: We appreciate the reviewer's suggestion. We have revised this section by simplifying the equations mentioned and adding equivalent circuit models to the figures, which enhances the clarity of the chapter for non-specialist readers. The modifications are detailed below:

    “During Phase II, cellular viscoelastic deformation occurs within the microchannel, where the measured deformation length Lp(t) adheres to a power-law rheological model. This dynamic behavior is governed by intrinsic cellular viscoelasticity, with core parameters including Young's modulus and the power-law exponent. Subsequently, deformation data Lp(t) from Phase II is utilized to generate theoretical deformation curves Lcell(t) via the power-law rheological model. Finally, intrinsic mechanical parameters Young's modulus and power-law exponent are determined through least-squares minimization. “

    1.    A comparative analysis of microfluidic impedance detection with other biosensing technologies (e.g., fluorescence-based assays, PCR, ELISA) would enhance the review. A side-by-side comparison of advantages, limitations, and performance metrics would help readers better understand the unique value proposition of this technology. The authors can consider adding some tables for the comparative analysis.

    Reply 2: We sincerely appreciate the reviewer's valuable suggestions. In response to these comments, we have conducted a comprehensive comparative analysis between microfluidic impedance sensing and other biosensing technologies (e.g., fluorescence-based detection, PCR, ELISA), as detailed below:

    Table 1. Comparison of microfluidic impedance detection technology with three other detection methods.

    Category

    Microfluidic Impedance Detection

    Fluorescence Detection[3,4]

    PCR[5,6]

    ELISA[7,8]

    Principle

    Label-free electrical detection based on dielectric property differences of cells/particles

    Fluorescent dye labeling with optical signal acquisition

    Nucleic acid amplification and fluorescence quantification

    Antigen-antibody binding with enzymatic chromogenic reaction

    Advantages

    Label-free operation, real-time dynamic monitoring, miniaturized equipment, low sample consumption, simplified workflow

    High sensitivity (single-molecule level), multi-channel parallel detection, mature technology

    Ultra-high sensitivity (aM–fM), exceptional specificity, quantitative capability

    High specificity, high throughput, standardized protocols, commercial maturity

    Limitations

    Susceptible to environmental interference, limited multi-target detection capability, complex chip design requirements

    Label-dependent (increased cost/time), photobleaching issues, expensive instrumentation

     Thermal cycler dependency, complex instrumentation, contamination risks, non-nucleic acid targets undetectable

    Antibody quality dependency (potential cross-reactivity), lengthy procedures, limited small-molecule detection

    Detection time

    Minutes(real-time)

    Minutes/hours

    1–3 hours

    3–24 hours

    Sensitivity

    pg/mL level

    fg/mL level

    aM–fM level 

    pg/mL level

    Throughput

    Medium–High (chip-dependent)

    Medium

    Medium

    High

    Cost

    Low equipment cost, high chip development cost

    High reagent cost (labels), high equipment cost

    High equipment/reagent costs

    Moderate equipment/reagent costs

    Applications

    Point-of-care testing, cellular analysis

    Single-molecule detection, imaging analysis

    Pathogen detection, gene expression

    Protein biomarkers, clinical diagnostics

    1.    The discussion on AI and machine learning integration is limited. While the article mentions AI-driven algorithms as a future direction, it does not provide detailed insights into how these technologies are currently being used to enhance data analysis and interpretation in microfluidic impedance systems.

    Reply 3: We are grateful to the reviewer for this insightful suggestion. In accordance with the comment, we have elaborated on how the AI-driven algorithms specifically reinforce data analysis and interpretation in microfluidic impedance systems. The detailed enhancements are presented below:

    “3) Artificial Intelligence (AI) Systems and data processing strategies: AI and machine learning (ML) are increasingly being used in the data analysis and processing of microfluidic impedance chip systems [73–75]: Convolutional Neural Networks (CNNs) automatically extract spatial features (e.g., cellular morphology, impedance distribution), where local patterns are captured through convolutional layers while dimensionality reduction and noise resistance are achieved via pooling layers. Raw images/signals are directly processed to identify cell-type differences, and multi-source parameters (temperature/pH) are fused to enable real-time sensor drift correction. Recurrent Neural Network variants (GRU/LSTM) model temporal dynamics (e.g., impedance changes during cell division, biomarker fluctuations in sweat) through gating mechanisms, with long-term dependencies captured to predict physiological trends. These are combined with CNNs to form hybrid architectures (CNN-GRU) for spatiotemporal feature co-analysis. These methods further enhance the ability of impedance signals to predict cell behavior and response, accurately predict future dynamic changes of cells through real-time and historical data and greatly enhance the diagnostic performance of microfluidic impedance analysis technology. In addition, AI algorithms can also be used to control microfluidic impedance systems, form intelligent feedback mechanisms, adjust experimental conditions (such as flow rate, measurement frequency, etc.) in real time, perform high-precision measurements of biological samples of different sizes and types, and achieve closed-loop control and adaptive adjustment. Furthermore, the AI method of integrating impedance analysis with multimodal data fusion such as optical imaging, biosensors, and Raman spectroscopy analysis is expected to further improve the sensitivity and specificity of microfluidic chips in diagnosis and detection.

    4) AI-driven impedance microfluidic chip design: With the development of AI intelligent design microfluidic chips, as well as artificial intelligence expert models based on large language models such as ChatGPT[74,76], the development of impedance microfluidic technology chip design AI and expert models will become the key to the next generation of impedance microfluidic technology. This will greatly lower the threshold for the application of impedance microfluidic chips and make the technology accessible to a wider range of biological research fields.”

    1. The review could include more real-world validation and case studies to demonstrate the practical impact of microfluidic impedance detection. Examples of its use in real-world settings would provide valuable insights into its scalability and challenges.

    Reply 4: We extend our sincere appreciation for the reviewer's valuable comment. Following thorough consideration of the comments, we have discussed more research studies on the application of microfluidic impedance sensing technology to clinical specimens.

    “For practical sample detection, Silva et al. [71] developed an impedance-based microfluidic electronic tongue. Multi-sensing units coated with nanostructured thin films were integrated, and chemometric methods including principal component analysis and partial least squares-discriminant analysis were employed. Accurate discrimination of authentic coconut water samples (classification accuracy >90%) was achieved, along with high-precision prediction of physicochemical parameters such as soluble solid content and total titratable acidity. Xue et al. [72] developed a copper nanoparticle-enhanced microfluidic flow injection analysis system. Electrochemical impedance spectroscopy was employed to reveal a 20-fold lower charge transfer resistance at copper nanoparticle-modified multi-walled carbon nanotube screen-printed carbon electrodes versus standard counterparts. Microfluidic parameters were optimized and this modified electrode was utilized to achieve highly sensitive dopamine (DA) detection (LOD = 0.33 nM). Across three authentic sample matrices – artificial cerebrospinal fluid, fetal bovine serum, and human urine – recovery rates of 96.5% to 103.8% were demonstrated, proving its practical utility. “

    1. The article briefly mentions technical bottlenecks (e.g., signal noise suppression, device miniaturization) but does not delve deeply into the practical challenges of implementing the technology in real-world settings. A more detailed exploration of these challenges would provide a more comprehensive understanding of the barriers to widespread adoption of this technology.

    Reply 5: We express our sincere gratitude to the reviewer for this constructive suggestion. Through rigorous deliberation, we have expanded the discussion on research challenges in accordance with the comment, as substantiated in the following modifications:

    “1) Signal Noise and Interference Suppression: Signal noise represents a critical challenge affecting data accuracy and system reliability during microfluidic impedance measurements. Due to the compact size and high sensitivity of microfluidic devices, susceptibility to external electromagnetic interference exists, particularly in open measurement environments. Furthermore, device-intrinsic parasitic capacitance, poor contacts, and electrode contamination introduce additional noise, degrading signal quality. Differential measurement techniques are typically employed to effectively suppress common-mode noise through differential signal extraction for automatic noise balancing. Shielding materials and electromagnetic isolation methods, such as copper foil tapes, are utilized to minimize external interference. Signal processing approaches including filtering algorithms and adaptive noise cancellation algorithms have been implemented for post-processing raw signals to enhance signal quality and stability. Although numerous studies attempt to model and analyze impedance data using statistical methods and machine learning techniques, limitations persist in generalization capabilities and practical applicability due to insufficient data volume and sample imbalance. Particularly, machine learning algorithms require extensive annotated datasets for training, which are often unavailable in real-world scenarios. Poor cross-platform adaptability further restricts large-scale deployment.

    2) Device Miniaturization, Integration, and Standardization: While microfluidic technology inherently offers miniaturization advantages, achieving further size reduction becomes complex when integrating multi-channel, multi-parametric detection systems. Conventional microfluidic systems typically rely on single electrode arrays for impedance measurements, whereas practical applications necessitate integrated multiple electrodes and intricate microchannel designs to accommodate diverse detection needs. Miniaturization of impedance analyzers remains challenging, impeding portable system development. Additionally, standardization deficiencies exist since most microfluidic chips are laboratory-customized via photolithography without uniform specifications for electrode dimensions/materials. This variability hinders standardized performance evaluation across methodologies.

    3) Biological Sample Complexity and Pre-processing Requirements: Microfluidic impedance technology finds significant applications in biomedical detection, particularly for rapid analysis of biological samples like blood and urine. However, the inherent heterogeneity of biological samples—including cell density, protein concentration, and ionic strength—directly affects impedance measurements. Sample pre-processing is crucial in practice. For instance, whole blood contains abundant cellular components and solutes; direct measurement may cause signal confusion due to conductivity differences between cell membranes and solutions. Accuracy enhancement often requires pre-processing steps such as dilution, centrifugation, or filtration. Consequently, designing integrated pre-processing zones (e.g., filtration/separation units) within microfluidic chips is essential.”

Reviewer 2 Report

Comments and Suggestions for Authors

The authors in the manuscript with ID micromachines-3640413 investigated the microfluidic impedance detection technology through the integration of microscale fluid manipulation and bioimpedance spectrum analysis , allowing the  real-time detection of biological samples. The microfluidic impedance analysis technology can be applied to organ-on-a-chip , blood detection, microbial detection and tumors detection.

This manuscript synthesizes the (i) basic principles of microfluidic impedance sensing technology,(ii) the modeling methods for optimization strategies of cellular equivalent circuit systems, and (iii)  recent research and progress in biological sensing applications.

The work is well structured into several chapters such as:  (1)Introduction ; (2)  Theory of Microfluidic Impedance Analysis and (3) Application of Microfluidic Impedance Analysis.

In the first part of the paper, the traditional detection methods are compared with point-of-care testing (POCT). They  highlighted the  disadvantages of traditional methods ( PCR or ELISA) such as operational complexity, expensive equipment, high specificity and time-consuming procedures in comparison with POCT.   One of the first advantage  of POCT is the elimination of complex labotatory processing and time and raw materials consuming by miniaturisation at micro scale of entire laboratory into a chip called “"lab-on-a-chip". The  point-of-care testing, and on-site analysis is possible by integration of microfluidics with impedance sensing technologies allow to biological detection, from single-cell to organ-level analyses.

The second part included a) Microchannel Flow Characterization; (b) Cell Model and Cell Deformation Model; (c) Microfluidic Impedance Static and Dynamic System Design and electrode design.

The third section contains a comprehensive summary of applications f Microfluidic Impedance Analysis in biological sensing.

In the fourth part, the authors summarised and concluded that several improvements and challenges remain: (i) Standardization and manufacturing simplification must be prioritized; (ii) Anti-interference capability and portability require targeted improvements; (iii) Intelligent data processing requires strategic development ; (iv) Multimodal integration represents a transformative direction; (v) Industry-academia collaboration serves as the implementation catalyst.

The manuscript is well-written, it has adequate and relevant references, and presented the outstanding outcome in the field of Advance microfluidic impedance detection technology . The review is of interest to all researchers  in the field of microfluidic technology applied  in bioengineering. Also, the work opens new horizons regarding advanced  microfluidic impedance detection technologies,  and has a complex approach that involves the interdisciplinarity between medicine-biology and physics.

As a result, I agree with the publication of this review, with small corrections:

Line 770 : Please complete the references [8]  [23] according with MDPI template, for instance:

[8] Shen, Y., Chen, Z., Zhang, Y., & Hou, X. (2023). Liquid interfacial tension design of building new concept materials. Matter6(8), 2506-2508, https://doi.org/10.1016/j.matt.2023.05.013

Line 806 : Please complete the references [8]  [23] according with MDPI template, for instance:

[23] Kim, B., Yao, W., Rhie, J. W., & Chun, H. (2022). Microfluidic potentiometric cytometry for size-selective micro dispersion analysis. BioChip Journal16(4), 471-479, https://doi.org/10.1007/s13206-022-00083-y

Author Response

  1.    Line 770 : Please complete the references [8]  [23] according with MDPI template, for instance:

Reply 1: We acknowledge the reviewer's meticulous scrutiny and have reformatted the references in full compliance with the journal's guidelines as per the comments. The detailed enhancements are presented below:

“16. Shen, Y.; Chen, Z.; Zhang, Y.; Hou, X. Liquid Interfacial Tension Design of Building New Concept Materials. Matter 2023, 6, 2506–2508, doi:10.1016/j.matt.2023.05.013.

  1. Kim, B.; Yao, W.; Rhie, J.W.; Chun, H. Microfluidic Potentiometric Cytometry for Size-Selective Micro Dispersion Analysis. BioChip J2022, 16, 471–479, doi:10.1007/s13206-022-00083-y.“

Reviewer 3 Report

Comments and Suggestions for Authors

See attachment for my feedback

Author Response

  1.    Section 2, line 111: please add a short introduction to this section, describing what is discussed in the subsections 2.1-2.4.

Reply 1: We appreciate the reviewer's input. In accordance with the suggestion, we have inserted a concise preamble to distill the subsequent content, with specific modifications detailed below:

“This chapter establishes a theoretical framework for microfluidic impedance analysis, spanning three domains: hydrodynamics, cellular equivalent circuit modeling, and system design. Hydrodynamic characteristics and microchannel flow behaviors are first elucidated. Equivalent circuit modeling methods for detecting cellular electro-mechanical properties via impedance measurements are subsequently introduced. Design principles for static and dynamic microfluidic impedance detection systems are comprehensively detailed, encompassing electrode configurations and accuracy enhancement strategies through electric field optimization, hydrodynamic focusing, and signal compensation. Progressing systematically from microscopic mechanisms to macroscopic implementations, this hierarchical framework provides the technical foundation for multidimensional single-cell characterization and precision sensing.”

  1.  Section 2, line 114: do the authors refer to micrometer-sized cross-sectional dimensions (width and height/depth), or the length dimensions of the channels? Please clarify.

Reply 2: We appreciate the reviewer's query. We recognized that the original statement failed to specify that the 'micron-scale dimensions' of the microfluidic chip refer exclusively to the channel width and height/depth. This clarification has been incorporated into the revised text, as detailed below:

“The essence of microfluidics lies in constructing miniaturized fluidic networks, where micron-scale channels (specifically width and height/depth dimensions within micrometer range) are precisely fabricated via microfabrication techniques (e.g., soft lithography, laser ablation, or injection molding). ”

  1.    Section 2, lines 115, 171 / eq.(2):is acronym Z * total not better than Z * enterity? If ‘enterity’ is to be used, please changed into ‘enterily’.

Reply 3: We express our gratitude for the reviewer's suggestion. In response to this recommendation, we have modified the equation.

(2)

  1.   Captions Figs. 2-6: shouldn’t correct copyright sentences (from publishers) not be included? Please check/adapt.

Reply 4: We sincerely appreciate your vigilance regarding copyright statement compliance. Profoundly grateful for this critical reminder, we have formalized the requisite corrections as substantiated below:

Figure 2. (e) Equivalent circuit model of the background fluid. (Reprinted with permission from [24], published by AIP Publishing, 2025); (f) equivalent circuit model at the orifice. (Reprinted with permission from [25], published by Elsevier, 2023).

Figures 2-6 have been modified accordingly.”

  1.    Section 3, line 351: please add a short introduction to this section, describing what is discussed in the subsections 3.1-3.3.

Reply 5: We appreciate the reviewer's input. In accordance with the suggestion, we have inserted a concise preamble to distill the subsequent content, with specific modifications detailed below:

“This chapter focuses on the diversified applications of microfluidic impedance detection technology in fields including tumor diagnostics, hematological analysis, organ-on-a-chip systems, and microbial detection. The broad application prospects spanning molecular to organ levels are comprehensively demonstrated, highlighting its significant translational value and potential from fundamental research to clinical practice. Current technical limitations are critically discussed. As systematically compared in Table 1, this study delineates the differential characteristics between Microfluidic impedance detection and three conventional detection methodologies.”

  1.    Section 4: the outlook as described appears correct, however, it would be of added value if the authors make it more ‘concrete’, for example by adding conceptual ideas (sketches) of a unified chip design, or what a design should fulfill.

Reply 6: We express sincere appreciation for the reviewer's valuable insights. Pursuant to the recommendations, we have comprehensively revised this technical segment and integrated a preliminary schematic diagram to enhance conceptual clarity. The specific modifications are substantiated as follows:

“Signal Noise and Interference Suppression: Signal noise represents a critical challenge affecting data accuracy and system reliability during microfluidic impedance measurements. Due to the compact size and high sensitivity of microfluidic devices, susceptibility to external electromagnetic interference exists, particularly in open measurement environments. Furthermore, device-intrinsic parasitic capacitance, poor contacts, and electrode contamination introduce additional noise, degrading signal quality. Differential measurement techniques are typically employed to effectively suppress common-mode noise through differential signal extraction for automatic noise balancing. Shielding materials and electromagnetic isolation methods, such as copper foil tapes, are utilized to minimize external interference. Signal processing approaches including filtering algorithms and adaptive noise cancellation algorithms have been implemented for post-processing raw signals to enhance signal quality and stability. Although numerous studies attempt to model and analyze impedance data using statistical methods and machine learning techniques, limitations persist in generalization capabilities and practical applicability due to insufficient data volume and sample imbalance. Particularly, machine learning algorithms require extensive annotated datasets for training, which are often unavailable in real-world scenarios.

Device Miniaturization and Standardization: While microfluidic technology inherently offers miniaturization advantages, achieving further size reduction becomes complex when integrating multi-channel, multi-parametric detection systems. Conventional microfluidic systems typically rely on single electrode arrays for impedance measurements, whereas practical applications necessitate integrated multiple electrodes and intricate microchannel designs to accommodate diverse detection needs. Miniaturization of impedance analyzers remains challenging, impeding portable system development. Additionally, standardization deficiencies exist since most microfluidic chips are laboratory-customized via photolithography without uniform specifications for electrode dimensions/materials. This variability hinders standardized performance evaluation across methodologies.

Biological Sample Pre-processing: Microfluidic impedance technology has significant applications in biomedical detection, particularly for rapid analysis of biological samples like blood and urine. However, the inherent heterogeneity of biological samples, including cell density, protein concentration, and ionic strength—directly affects impedance measurements. Sample pre-processing is crucial in practice. For instance, whole blood contains abundant cellular components and solutes; direct measurement may cause signal confusion due to conductivity differences between cell membranes and solutions. Accuracy enhancement often requires pre-processing steps such as dilution, centrifugation, or filtration. Consequently, designing integrated pre-processing zones (e.g., filtration/separation units) within microfluidic chips is essential.”

7.Grammar (not complete!) 

  • Throughout manuscript:define all acronyms (only) at first point of use (PDMS is defined in lines 54 and 118 etc., PBS in line 212, AC in lines 135/261 etc., IFC and EIS (lines 337/338), RBC (line 443).
  • Authors, line 4: the word ‘and’ should be before the last author (Yigang Shen), please adapt.
  • Introduction, line 35:please add a space between ‘methods’ and ‘(such…)’.
  • Lines 147, 152, 153, 154:incorrect of subscripts (should be used), please adapt.
  • Lines 295, 296:incorrect of space before ‘comma’.
  • Lines 382, 383:“Another side. Luan et al. [29]…” is incorrectly phrased.
  • Lines 388-390:it should be “To solve this problem, by simultaneously…capacitance Feng et al. [30] achieved…”

Reply 7: We extend our sincere gratitude to the reviewer for identifying these oversights. Our initial submission contained inadvertent lapses in language conventions and formatting protocols. Informed by these critical observations, we have meticulously rectified all flagged discrepancies
